# Cellular Immunology of Myocarditis: Lights and Shades—A Literature Review

**DOI:** 10.3390/cells13242082

**Published:** 2024-12-17

**Authors:** Cristina Vicenzetto, Andrea Silvio Giordani, Caterina Menghi, Anna Baritussio, Federico Scognamiglio, Elena Pontara, Elisa Bison, Maria Grazia Peloso-Cattini, Renzo Marcolongo, Alida Linda Patrizia Caforio

**Affiliations:** 1Cardiology, Department of Cardiac Thoracic Vascular Sciences and Public Health, University of Padova, Via Giustiniani 2, 35128 Padova, Italy; 2Cardioimmunology Laboratory, Department of Cardiac Thoracic Vascular Sciences and Public Health, University of Padova, Via Giustiniani 2, 35128 Padova, Italy

**Keywords:** myocarditis, immune system, immunosuppressive therapy, autoimmune disease, systemic immune-mediated disease

## Abstract

Myocarditis is an inflammatory disease of the myocardium with heterogeneous etiology, clinical presentation, and prognosis; when it is associated with myocardial dysfunction, this identifies the entity of inflammatory cardiomyopathy. In the last few decades, the relevance of the immune system in myocarditis onset and progression has become evident, thus having crucial clinical relevance in terms of treatment and prognostic stratification. In fact, the advances in cardiac immunology have led to a better characterization of the cellular subtypes involved in the pathogenesis of inflammatory cardiomyopathy, whether the etiology is infectious or autoimmune/immune-mediated. The difference in the clinical course between spontaneous recovery to acute, subacute, or chronic progression to end-stage heart failure may be explained not only by classical prognostic markers but also through immune-pathological mechanisms at a cellular level. Nevertheless, much still needs to be clarified in terms of immune characterization and molecular mechanisms especially in biopsy-proven myocarditis. The aims of this review are to (1) describe inflammatory cardiomyopathy etiology, especially immune-mediated/autoimmune forms, (2) analyze recent findings on the role of different immune cells subtypes in myocarditis, (3) illustrate the potential clinical relevance of such findings, and (4) highlight the need of further studies in pivotal areas of myocarditis cellular immunology.

## 1. Introduction

Myocarditis is an inflammatory disease of the myocardium, diagnosed through well-established histological, immunological, and immunohistochemical criteria. It presents with a broad spectrum of etiologies, clinical manifestations, and outcomes [1,2]. Acute myocyte damage may trigger the activation of both innate and adaptive immune responses, leading to an inflammatory response. In most patients, the immune reaction is eventually downregulated, and the myocardium recovers. In a sizable portion of cases, however, persistent inflammation leads to ongoing myocyte damage, resulting in ventricular dilation, reduced contractility, and end-stage heart failure [3,4]. When associated with myocardial dysfunction, biopsy-proven myocarditis is termed inflammatory cardiomyopathy [1], a complex and clinically relevant disorder.

In recent years, extensive research efforts have been made to elucidate the underlying immunopathogenesis of myocarditis, employing a variety of methodologies, including animal models [5,6] and human studies on patients and healthy controls [7,8,9]. These investigations aimed to provide a comprehensive understanding of how both innate immune cells (e.g., macrophages and dendritic cells) and adaptive immune cells (e.g., T cells and B cells) initiate and amplify the immune response [10,11,12]. However, despite these advances, many aspects, particularly the immune characterization and molecular mechanisms underlying biopsy-proven autoimmune myocarditis and inflammatory cardiomyopathy, remain poorly understood.

The aims of this review are to (1) describe inflammatory cardiomyopathy etiology, with a focus on immune-mediated/autoimmune forms, (2) analyze recent findings on the role of different immune cells subtypes in myocarditis, (3) illustrate the potential clinical relevance of these findings, and (4) highlight the need for further studies in pivotal areas of myocarditis cellular immunology.

## 2. Clinical Presentation of Myocarditis and Inflammatory Cardiomyopathies

Myocarditis encompasses a heterogeneous spectrum of clinical manifestations, ranging from mild to severe forms. This variability complicates diagnosis, risk assessment, and therapeutic management [2]. The majority of the cases of clinically suspected infarct-like cases occur in men (60–80%), with a median age of presentation between 30 and 45 years [13,14]. However, the incidence of myocarditis, especially with elusive clinical presentations such as arrhythmias or chronic heart failure, is difficult to assess due to the inconsistent use of the diagnostic gold standard, endomyocardial biopsy (EMB).

In fact, myocarditis may even be paucisymptomatic, with a slow and insidious course, thereby often leading to delayed diagnosis. Conversely, it may present with a sudden onset of unexplained cardiac signs and symptoms or progress rapidly to a fulminant form [2,15]. Commonly reported symptoms include chest pain (82% to 95% of cases), fever (58–65% of cases), dyspnea (19–49% of cases), and syncope (5–7% of cases) [16,17]. In approximately 7% to 12% of acute myocarditis cases, the onset is fulminant, characterized by cardiogenic shock (CS) or acute heart failure (HF) and left ventricular (LV) dysfunction, with or without malignant ventricular arrhythmias and/or conduction abnormalities [18,19].

Clinical manifestations may also mimic those of other cardiomyopathies, including Takotsubo syndrome (TS) [20,21] and arrhythmogenic right ventricular cardiomyopathy (ARVC). In a non-negligible proportion of cases, ARVC patients may present with acute chest pain episodes and elevated myocardial enzyme levels, frequently with preserved biventricular systolic function, during so-called ‘hot phases’, which have been observed in 5% of a previously reported ARVC cohort [22,23]. This is even more relevant considering the newly identified phenotypes of arrhythmogenic cardiomyopathy (ACM), including the “left-dominant” and “biventricular” disease subtypes [24], in which a phenotypical and clinical overlap with inflammatory cardiomyopathy should be carefully investigated, in order to promptly define a correct diagnostic and therapeutic patient work-up. Since histological evidence of inflammatory infiltrates in ARVC patients has been provided since the 1990s, multi-parametric assessment of myocarditis in the context of ACM, especially during the so-called “hot phases”, is encouraged [25].

## 3. Challenges in Diagnosis: The Role of Endomyocardial Biopsy and Imaging Techniques

EMB is the diagnostic gold standard for myocarditis. According to the 2013 position statement of the European Society of Cardiology Working Group on Myocardial and Pericardial Diseases, EMB should not be restricted to hemodynamically or electrically unstable patients but should instead be considered for any clinically suspected myocarditis case where a definitive etiological diagnosis could impact the outcome. This has been reinforced by the latest consensus statement by the three leading international HF societies [19]. EMB should be performed following the exclusion of other potential cardiac or extracardiac conditions that could explain the symptoms and imaging findings, particularly coronary artery disease, which can be ruled out through invasive coronary angiography or coronary CT, according to the patients’ pretest probability of relevant coronary atherosclerosis [2,26,27]. The EMB examination is based on conventional histopathological analysis according to the Dallas criteria [28,29], supplemented by immunohistochemical characterization of the inflammatory infiltrate and polymerase chain reaction (PCR) detection of infectious agents. The type of inflammatory infiltrate—whether eosinophilic, polymorphous, lymphocytic, or granulomatous—plays a critical role in both prognostic stratification and therapeutic decision-making [2,3,26].

When EMB is not initially performed during clinical evaluation—particularly in cases where any markers of severe prognosis are present and specific etiological treatment is not required—the 2013 ESC criteria allow for a diagnosis of clinically suspected myocarditis [2,19]. This diagnosis is largely based on ruling out coronary artery disease through coronary angiography and identifying myocarditis typical findings on CMR imaging.

Over the past two decades, CMR has emerged as a reliable and accurate non-invasive diagnostic technique to support clinical suspicions of myocarditis, due to its ability to provide volume quantification, contractility assessment, and myocardial tissue characterization [30,31]. In 2009, the original Lake Louise criteria (LLC) were established to enhance CMR diagnostic accuracy for suspected myocarditis through uniform protocols. These criteria included (1) global or regional myocardial systolic dysfunction, (2) myocardial edema, and (3) myocardial hyperemia or increased vascular permeability, as indicated by early (EGE) or late gadolinium enhancement (LGE) on CMR [32]. The diagnosis of myocarditis required at least two of the three aforementioned criteria, with one being either myocardial edema or myocardial LGE. To improve diagnostic accuracy, the latest revisions to the LLC have integrated new mapping techniques and now require both myocardial edema (one T2-based criterion) and non-ischemic myocardial injury (one T1-based criterion) to be present in order to raise suspicion of myocardial inflammation [33,34] (Figure 1).

While CMR is valuable in various clinical scenarios, especially in distinguishing between coronary-ischemic and inflammatory myocardial damage, it does not provide information about the underlying etiology or the histological subtype of myocarditis [15].

Furthermore, the prognostic role of LGE in myocarditis is still open to debate. While studies on non-ischemic cardiomyopathy found LGE to be quantitively associated with worse outcomes [37], a recent single-center study on 207 clinically suspected or biopsy-proven myocarditis patients showed that higher biventricular systolic function and a greater extent of LGE on CMR at diagnosis were associated with better outcomes when assessed at any follow-up point. Conversely, larger biventricular volumes, CMR-based dilated cardiomyopathy (DCM) features, and the presence of an ischemic LGE pattern at diagnosis were predictors of worse outcomes [38].

Regarding nuclear imaging techniques, positron emission tomography with 2-deoxy-2-fluoro-D-glucose (FDG-PET) can be used to evaluate the inflammatory activity of the heart and monitor treatment responses specific conditions, such as cardiac sarcoidosis [39,40,41]. However, data regarding the use of FDG-PET in myocarditis evaluation remain limited. Only a few case reports and series have documented FDG findings in viral myocarditis [42,43,44,45,46], eosinophilic myocarditis [43], and GCM [44,45].

## 4. Etiology of Myocarditis: Viral and Toxic Causes

The etiopathogenesis of myocarditis is complex, primarily categorized as either viral or autoimmune/immune-mediated [2]. Myocarditis can result from various infectious agents, with viruses being the most common culprits, though bacteria and parasites may also play a role. While viral infections dominate in Western countries, Central and South America present a higher incidence of Chagas disease, which is caused by the protozoan Trypanosoma Cruzi [15]. Notably, around 27% of patients may present with multiple viral infections affecting the myocardium [4].

Viral myocarditis can be categorized based on the viral tropism. Viruses that are primarily cardiotropic, such as adenoviruses and enteroviruses, directly target the myocardium, while others like parvovirus B19 (PVB19) are vasculotropic and may persist lifelong. Lymphotropic viruses, including cytomegalovirus (CMV) and Epstein–Barr virus (EBV), target lymphatic tissues, and certain strains like Influenza A and B can exert cardiotoxic effects on the myocardium [4]. Accurate diagnosis of viral myocarditis is essential and should be based on PCR testing from myocardial tissue rather than serological markers, which may not reliably indicate current infection [4,47]. Given the complexity of viral pathogens, treatment strategies should be multidisciplinary, with close collaboration with infectious disease specialists to ensure tailored antiviral therapy, especially in cases of chronic or recurrent infections [19,48].

Toxic causes of myocarditis are a minority, occurring either as hypersensitivity myocarditis, which is unrelated to drug dosage, or as dose-dependent direct cardiac toxicity. Among these less common causes is mesalazine, an established first-line treatment for inflammatory bowel disease (IBD) and mainstay therapy for mild-to-moderate ulcerative colitis (UC) [49]. While mesalazine has been associated with myocarditis, with a reported incidence as high as 0.3% and potentially fatal outcomes [50], data from a single-center experience and a literature review suggest that, in the absence of EMB-based confirmation, the true incidence may be overestimated [51].

An immune-mediated pathological reaction may arise from clear toxic myocardial damage. A clear example of a toxic myocarditis with an underlying autoimmune mechanism is myocarditis induced by immune checkpoint inhibitors (ICIs), which warrants particular attention due to the relevant morbidity and mortality [52]. Over the past decade, ICIs have significantly improved outcomes for cancer patients. However, ICIs can also trigger a wide range of potentially life-threatening immune-related adverse events (irAEs) including fulminant myocarditis [53].

## 5. Focus on Immune-Mediated Myocarditis and Inflammatory Cardiomyopathy

The role of autoimmunity in myocarditis is well established, occurring either as post-infectious immune-mediated myocardial damage or as a primary organ-specific autoimmune disease [2]. Additionally, non-infectious autoimmune myocarditis may occur in various systemic immune-mediated diseases (SIDs) [54], which include autoimmune and autoinflammatory diseases affecting at least two organ systems [55], such as systemic lupus erythematosus (SLE), rheumatoid arthritis (RA), systemic sclerosis (SSc), and mixed connective tissue disease (MCTD). Myocarditis is a hallmark of SIDs, often associated with worse prognoses and necessitating an intensified immunosuppressive regimen [39,56,57].

In both organ-specific myocarditis and systemic immune-mediated myocardial damage, the role of humoral immunity is well established. Research dating back to the late 1980s and early 1990s reported the presence of anti-heart autoantibodies (AHAs) [58,59,60,61] in cases of acute and chronic myocarditis or DCM [62,63,64]. These biomarkers are present in 60–80% of patients with biopsy-proven organ-specific autoimmune myocarditis/inflammatory cardiomyopathy across its full spectrum of clinical presentations, including fulminant, acute, subacute, chronic heart failure, pseudo-infarction, and arrhythmic presentation [3,62,65], and their presence is correlated with poor prognosis [66]. Furthermore, their detection in asymptomatic relatives of patients with idiopathic DCM may serve as a predictive marker for disease development [65,67]. AHA antigens are the α and β isoforms of the cardiac myosin heavy chain and are therefore considered cardiac-specific autoantigens [59].

AHAs and anti-intercalated disc autoantibodies (AIDAs) serve as disease-specific markers of immune-mediated myocardial damage also in the context of SIDs, such as systemic sclerosis [68] and sarcoidosis with cardiac involvement. [69]. Future studies are warranted to clarify whether or not AHAs and AIDAs play a direct pathogenic role in systemic immune-mediated myocardial damage, as suggested in organ-specific myocarditis [70]. Other autoimmune biomarkers, such as anti-Desmoglein-2 antibodies, have been identified in myocarditis and various other cardiac and non-cardiac immune-mediated diseases. These biomarkers not only correlate with specific disease features and prognostic markers but also suggest a potentially pivotal role in disease pathogenesis [71,72,73].

## 6. Interplay of Genetic Predisposition and Autoimmunity

Autoimmune diseases typically arise from the interaction between genetic predisposition and environmental triggers. This interplay results in immune dysregulation and a failure to recognize self-antigens, ultimately leading to a loss of tolerance. Similarly, in autoimmune myocarditis, numerous studies have sought to clarify the relationship between environmental factors, genetic background, and disease development [74,75,76]. Genetic factors may significantly influence both susceptibility to myocarditis and its clinical evolution, particularly in patients with severe left ventricular (LV) dysfunction who may progress to DCM [77].

Genetic polymorphisms in the major histocompatibility complex (MHC) genes can affect antigen binding affinities, with specific MHC genes linked to an increased risk of certain autoimmune diseases [78]. In humans, alleles such as HLA (human leukocyte antigen)-DR4, HLA-DR12, and HLA-DR15 have been associated not only with the development of myocarditis but also with a higher risk of progression to DCM [79,80]. Furthermore, recent transcriptomic studies have revealed a higher prevalence of HLA-DQ1 expression in patients with myocarditis compared to those to those without. In fact, transgenic mice expressing human HLA-DQ8 or HLA-DR4 have been shown to spontaneously develop fatal autoimmune myocarditis [5,81]. Besides HLA polymorphism, recent data suggest an overlap between certain genetic cardiomyopathies and myocarditis. A population-based cohort study that included 336 consecutive myocarditis patients, mainly with a clinically suspect diagnosis, revealed that pathogenic mutations in cardiomyopathy-related genes (i.e., pathogenic or likely pathogenic variants in genes related to specific cardiomyopathies) were present in 8% of the myocarditis cases and in less than 1% of healthy controls [74]. These genetic variants were detected in both genes associated with DCM, such as *TTN*, and those linked to ARVC, like *DSP*. Importantly, patients carrying these genotype-positive mutations showed a poorer prognosis and increased 5-year mortality rates [74].

## 7. Immunosuppressive Regimens for Myocarditis: Current Evidence and Future Challenges

Effective management of myocarditis requires a comprehensive therapeutic approach designed to address both cardiovascular complications and the specific underlying etiology, whether viral or autoimmune.

Immunosuppressive therapy (IS) is a cornerstone in the management of biopsy-proven (BP) autoimmune myocarditis, aiming to attenuate inflammation and prevent myocardial injury [2,82]. Typical immunosuppressive regimens involve corticosteroids combined with agents like azathioprine or cyclosporine over a six-month period. Alternatively, other immunosuppressive drugs such as mycophenolate mofetil or methotrexate may be used alongside steroids [82,83,84].

Evidence supporting the efficacy of IS for treating heart failure in lymphocytic virus-negative myocarditis mainly derives retrospective studies and meta-analyses [82,83,85,86]. A recent propensity-score-based study assessed the long-term safety and effectiveness of personalized IS therapy in 91 patients with BP immune-mediated myocarditis. The study found comparable survival rates and heart function at the 5-year follow-up in IS-treated patients with BP immune-mediated myocarditis, compared to 267 controls receiving only optimal medical therapy (OMT). Minor manageable adverse reactions occurred in just 13% of IS patients [35].

For other less common histological forms of myocarditis such as GCM [44], eosinophilic myocarditis [87], and cardiac sarcoidosis [88], data on the efficacy of IS come from retrospective observational registries, and further studies are needed for a complete characterization of the optimal types and duration of a tailored IS in these settings.

Despite advances in diagnostic techniques, a standardized treatment approach for myocarditis remains elusive, and individual responses to IS treatment vary, with some patients showing significant improvement, while others remain refractory to therapy [36]. This is primarily due to the unknown mechanisms governing host immune responses, which can either eliminate the virus and resolve inflammation or lead to persistent immune-mediated damage with or without viral clearance. Therefore, predictors of a favorable response to IS in myocarditis, including peripheral non-invasive biomarkers, are still under investigation.

## 8. Role of Different Immune Cell Populations and Cytokines in Myocarditis

Cardiomyocytes are the major cellular component in the heart, but many other cell types are present to allow proper cardiac functionality (Figure 2, Table 1). Among these are resident immune cells, such as macrophages. Resident monocytes, mainly CX3CR1+ and of embryonic origin, establish physical contact with neighboring cardiomyocytes. At basal conditions, these cells exert a tissue remodeling role. On the other hand, cardiac dysfunction induces cardiomyocytes to secrete monocytes recruiting chemokines. These recruited monocytes differentiate into CCR2+ macrophages, which are pro-inflammatory [89] and have been identified in EMB of myocarditis patients [90]. Indeed, in experimental autoimmune myocarditis (EAM) models, as well as in human EMB, macrophages are the predominant infiltrating cells, and their pro-inflammatory M1 polarization can be driven by mitochondrial fission [91]. In EAM models, the infiltration kinetics of classical monocytes, contributing to CCR2+MHCII+ macrophage compartment, peaked at 14 days of immunization, while the non-classical monocytes peaked at 21 days [92,93]. This double recruitment of CD11b+ monocytes resulted both in a strong pro-inflammatory signal in the first peak as well as a suppression of myosin heavy chain (MyHC)-specific Th17 T cell responses in the second peak through a disease-induced limiting IFN-γ-triggered negative feedback loop [93]. Autoreactive T cells recruited monocytes either directly, as in the case of Th17 cells, or by IL-3 secretion, which incites tissue macrophages to produce monocyte-attracting chemokines [92,94]. Recently, it has been demonstrated that monocyte recruitment can be blocked by siRNA silencing of CCR2 in EAM, leading to reduced myocardial inflammation [90].

Many other myeloid cells contribute to myocarditis onset/progression, including granulocytes and dendritic cells. Regarding granulocytes, neutrophil extracellular traps (NETs) have been identified in EMB samples from patients and EAM mice, and their inhibition can reduce inflammation, including in giant cell myocarditis, the most fatal form [95,96]. Moreover, eosinophils can strongly infiltrate the myocardium, as observed in transgenic mice overexpressing IL-5, leading to fatal eosinophilic myocarditis, which is one of the most aggressive forms of myocarditis in humans [97]. Eosinophils are not essential for myocarditis initiation, but they are fundamental in mediating DCM evolution through IL-4 secretion and a Th2 deviation [97,98].

Historically, myocarditis was defined as a T-cell-mediated diseases, but given the large amount of studies proving the role of nearly all inflammatory cell types in myocarditis development, nowadays myocarditis pathogenesis should be described as a state of general immune dysregulation, including both adaptive and innate immunity, with the latter playing a fundamental role in antigen presentation to T cells [9,99,100,101]. Among the key players, dendritic cells (DCs) are a heterogeneous type of professional antigen presenting cells that might derive from myeloid precursors, as well as from monocytes. Alterations in various DC subsets have been observed in the peripheral blood of myocarditis patients. In acute myocarditis, higher levels of DCs with a stronger expression of co-stimulatory proteins have been reported compared with healthy controls, suggesting a higher immunogenic state that might prime better T cells [7]. Conversely, in a mixed cohort of patients with suspected and biopsy-proven myocarditis, with DCM evolution, a strong reduction in plasmacytoid and myeloid dendritic cells has been described in peripheral blood, with a concomitant accumulation in patients’ myocardium [8]. These opposite results could be due to different patients’ enrollment criteria, as well as different panels for DC characterization, but they are indicative of a potential pathogenetic role of DCs in myocarditis, a hypothesis supported by several myocarditis mouse models. In fact, DCs can induce T lymphocytes and exacerbate the mouse myocardial inflammation through the glycoprotein Tenascin-C, which induces inflammatory cytokine expression and activation of Th17 cells via Toll-like receptor 4 [102]. Moreover, after EAM induction, type 2 conventional DCs (cDC2) have been reported to specifically present α-myosin and induce Th1 and Th17 cell differentiation [101]. The accumulation of cDC2 and plasmacytoid DCs in inflamed myocardial tissue and their immune-related pathway activation have been recently described by an integrated single-cell RNA sequencing analysis of two different EAM model gene expression data sets [103]. The generation of tolerogenic (t)DCs demonstrated the pathogenicity of DCs and opened new potential therapeutic options. Specifically, myosin-pulsed tDCs can ameliorate EAM by antigen-specific Treg cell stimulation, as they overexpress the long noncoding RNA MALAT1 [104,105]. Treg cell stimulation is fundamental to restore normal cardiac immunity, as these cells have been demonstrated to be reduced and less functional in DCM and acute myocarditis [9,106,107,108]. Moreover, human extracellular vesicles, isolated from media conditioned with human-heart-derived stromal/progenitor cells, can induce Treg cell differentiation and promote the secretion of anti-inflammatory cytokines, as IL-10, both in vitro and in EAM models [109].

Dysfunctional Treg cells are strictly related to the strong activation and increase in Th17 cells, which so far have been the most extensively studied cells and proved to play a pathogenic role in myocarditis/DCM; thus, the Th17/Treg ratio is in favor of Th17 cells in myocarditis [9]. This imbalance might be linked to increased miR-155 levels in inflamed hearts, since in EAM models, it could facilitate Th17 cell differentiation as well as Treg suppression, and its inhibition rebalances the Th17/Treg ratio [110]. This imbalance has also been targeted with small molecules, such as fenofibrate, and with antibodies against the Th17 cell cytokines, such as IL-17 and IL-23, leading to an improvement of cardiac inflammation in models [111,112,113]. The most relevant role played by Th17 cells is the contribution to DCM evolution; in particular, IL-23 and IL-6 signaling induce Th17 cells to differentiate and infiltrate the heart, and, in fact, their silencing blocks EAM onset/evolution [114,115,116]. IL-6 is a fundamental cytokine for myocarditis development, since EAM models IL-6^−/−^ are resistant to myocarditis development [117]. Recently, a temporal characterization of heart-infiltrating CD45+ cells in EAM mice showed that Th17 cells, overexpressing Hypoxia Inducible Factor (Hif)1α was the predominant T-cell population during the acute inflammatory phase, whereas Treg cells are detected at the subacute inflammatory phase, and γδ T cells releasing Il-17 are the main T-cell population observed at the myopathy phase [118]. IL-17A, produced by infiltrating Th17 cells, induces the production of monocyte-chemoattracting chemokines by cardiac fibroblasts to recruit inflammatory monocytes, underlining the fine immune cell cross-talk taking place in myocarditis evolution [119]. Moreover, Th17 cells further enhance their own differentiation through a positive feedback loop, since IL-17-A induces a heart-specific upregulation of IL-6, TNF-α, and IL-1 and promotes the recruitment of CD11b+ monocyte and Gr1+ granulocyte populations to the heart. Furthermore, IL-17A-deficient mice had reduced interstitial myocardial fibrosis [111]. The pathogenicity of Th17 cells has also been demonstrated in human myocarditis. A higher presence of a small noncoding RNA, i.e., hsa-Chr8:96, a homolog of the murine mmu-miR-721 produced by Th17 cells in EAM models and not in acute myocardial infarction, has been found in patients with acute myocarditis and biopsy-proven myocarditis [120]. Moreover, Th17 cells are found to be increased in the peripheral blood of suspected myocarditis/DCM patients with persistent heart failure and are detected also in EMB, correlating with higher levels of cardiac fibrosis. A proof of concept for the role of Th17 cells in disease progression can be obtained by evaluating Th17-associated cytokines in patients’ plasma: IL17-A is increased only after 6 months of disease onset, while IL-6 and TGF-β1 (cytokines relevant for Th17 cell differentiation) are increased at diagnosis. Th17 cell differentiation might be induced by cardiac myosin in human myocarditis, because human cardiac myosin S2 hinge region (S2-16 and S2-28) peptides act as endogenous ligands for Toll-like receptor 2, leading to an exaggerated response from CD14+ monocytes to secrete Th17-differentiating cytokines [9].

The relevance of other T helper subtypes has been more debated than Th17 cells, despite CD4+ cells being known to be fundamental to myocarditis pathogenesis, as the treatment of EAM rats with anti-CD4 antibodies blocks the development of the disease [100]. In a T cell receptor (TCR) transgenic mouse model specific for myosin heavy chain α (residue 614–629) spontaneously developing myocarditis, heart-infiltrating CD4+ T cells secrete IFN-γ and IL-17, indicating their Th1/Th17 phenotype. In particular, IFN-γ signaling is needed for spontaneous myocarditis development, while IL-17A, also in this model, has been linked to disease severity and DCM development [121]. The same findings have been described in in vitro stimulation of EAM mouse splenocytes with a recombinant fragment of cardiac myosin (1074–1646), obtaining lymphocytes secreting more IFN-γ, IL-6, and IL-17 than IL-4 [122]. This evidence underlines the importance of Th1 cells in myocarditis, even if their precise role is still debated. In classical EAM models, MyHC-α-specific Th cells more frequently differentiate towards IFN-γ-secreting cells [123]. Moreover, the transfer of CD4+ T cells specific to influenza hemagglutinin (HA) into transgenic mice expressing HA under the MyHC-α promoter gives rise to a cytotoxic Th1 phenotype, given the increased secretion of chemokines, i.e., Macrophage Inflammatory Protein (MIP)-1α and CCL5, which stimulates CD8+ cell migration [124]. On the other hand, Th2 lymphocytes, for the first time, have been implicated in the pathogenesis of EAM models developing eosinophilic or giant cell myocarditis, and the treatment with anti-IL-4 reduced the disease severity [125]. Conversely, in keeping with most of the studies on Th1 cells, another study with EAM rats showed that Th1 cytokines were increased in the acute phase, while they decreased during the recovery phase, when Th2 cytokines and IL-10 levels increased [126]. The clear role of Th2 cells is still to be clearly determined, but a recent study describe that, in advanced stages of EAM myocarditis, Bcl2-like protein (Bcl2L)12, by complexing the master regulator of Th2 differentiation GATA3, favors Th2 cells differentiation by IL-4 expression and blocks Th2 apoptosis inhibiting the expression of p53, leading to a Th2-mediate inflammation in the heart [127].

Other types of CD4+ lymphocytes, such as Th22 cells, have been reported to be increased in peripheral blood of DCM patients, indicating that a broader range of T helper cells might be involved in myocarditis/DCM pathogenesis [128]. Even if the clear pathogenetic mechanisms are not fully understood, this review has clearly described the double-edged sword role of CD4+ cells in both initiating and mediating recovery in myocarditis; this dual role could be due to TNF-α signaling, which promotes myocarditis development by activating cardiac endothelial cells to recruit T cells, but it can also trigger the activation-induced cell death pathway in cardiac-reactive T cells [129]. Moreover, a type of CD4+ cell that preferentially migrates to the heart has been described. These cells express hepatocyte growth factor receptor c-mesenchymal epithelial transition factor (c-Met) and chemokine receptors CXCR3 and CCR4 and present unique features, as they are able to secrete a mixture of cytokines from the different T helper cells described so far, such as IL-4/IL-13, IL-17, and IL-22, both in EAM models and acute myocarditis cases. c-Met+/CD4+ memory T cells have been identified in both inflammatory DCM and hereditary forms of other cardiomyopathies, suggesting that the immune system’s involvement should be considered even in cases where it is not the primary pathogenetic mechanism. On the other hand, c-Met+/CD8+ memory T cells are more specifically present in DCM. This could be due to the crucial role of CD8+ T cell responses in viral containment during viral infections [106]. Recently, an increased presence of exhausted CD8+ lymphocytes has been identified in both EMB and peripheral blood of patients with inflammatory DCM, with higher levels correlating with a worse prognosis in a combined cohort of inflammatory and non-inflammatory DCM cases [130]. Nonetheless, in models of T cell receptor (TCR) transgenic (Tg) mice specific to cardiac myosin heavy chain (MyHC)-α 334–352, CD4+ T cells have been shown to harbor a cytotoxic phenotype, since they express CD107a, IFN-γ, granzyme B, and natural killer cell receptors (NKG)2A and NKG2D [131].

The T cell compartment in myocarditis may develop distinct features due to expansion of cells equipped with a TCR specifically targeting myocardium. In various models, including in vitro studies, it has been shown that T cells often target the α isoform of MyHC, which is the predominant antigen detected in this context [121,123,124,131,132]. Interestingly, although α-MyHC constitutes only a small fraction of MyHC in the human heart, it has been evidenced that in EAM and in humans, medullary thymic epithelial cells, which are critical for central T cell tolerance, lack the expression of α-MyHC, leading to a defective control over α-MyHC autoantigens, allowing the increased presence of α-MyHC-T cells specifically in peripheral blood [99]. In addition to reduced immune tolerance, structural and immunological mimicry with viral and bacterial infections can contribute to the development of T cells targeting the myocardium, as streptococcal M protein-reactive T cells can target cardiac myosin [133]. Moreover, the evaluation of EMB from DCM patients reveals a preferential use of TCRVβ in infiltrating T cells, particularly in case of DCM with a viral etiology [134].

## 9. Future Perspectives and Clinical Applications: Targeted Immunosuppression

Among novel approaches to immune-mediated myocarditis, monoclonal antibodies represent a promising option, including rituximab, which acts against CD20+ B cells [135,136], and mepolizumab, which inhibits the binding of interleukin-5 (IL-5) to its receptors expressed on eosinophils, improving cardiac function [137,138,139].

IL-1β, a proinflammatory cytokine in the innate immune pathway that is crucial for host defense, represents another possible target for pharmacological intervention in myocardial inflammation. While neutralizing IL-1β has shown promise in reducing inflammation, interstitial fibrosis, and adverse cardiac remodeling in experimental animal studies [140,141,142], the ARAMIS (Anakinra versus placebo double-blind Randomized controlled trial for the treatment of Acute Myocarditis) phase 3 placebo-controlled trial did not show a net benefit in terms of in terms of heart failure episodes, chest pain, left ventricular ejection fraction < 50%, and ventricular arrythmias in patients with clinically suspected acute myocarditis of unspecified etiology presenting with chest pain and normal left ventricular (LV) function [143]. This is likely due to the low adverse event rate in the trial, and to the prevalent involvement of autoimmune, rather than autoinflammatory mechanisms in a sizable proportion of myocarditis cases. Further studies are needed to clarify the role of IL-1 receptor antagonists in patients with biopsy-proven immune-mediated acute myocarditis, considering the evidence supporting the role of innate immunity in myocarditis [144].

## 10. Conclusions

Myocarditis has been increasingly recognized as common cause of sudden cardiac death in young adults and heart failure overall.

Despite advancements in both experimental and clinical research, the immunological background of myocarditis remains only partially understood. Exploration of the specific cytokines and molecular pathways, both within myocardium and at the peripheral level, as well as the assessment of genetic predisposition, warrants further studies. Furthermore, the predictors of IS response in myocarditis are still under investigation.

The success of future trials for immunosuppressive treatments in myocarditis will depend also on immunophenotyping characterization of patients with myocarditis. This will help identify individuals with ongoing inflammation or abnormal immune responses, who are the most likely candidates to benefit from IS therapy.

## Figures and Tables

**Figure 1 cells-13-02082-f001:**
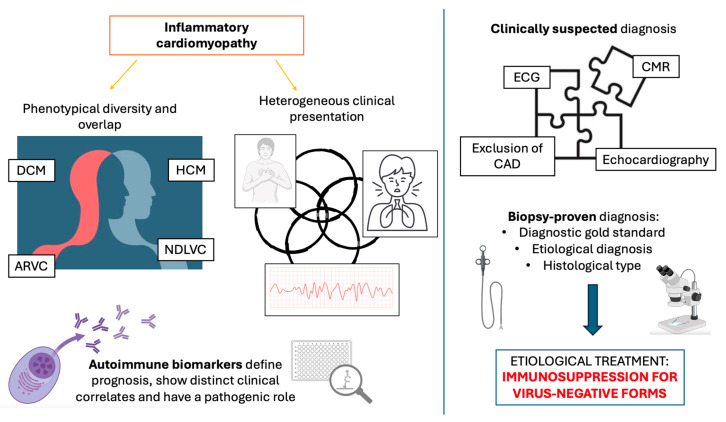
Overview of a guideline-based clinical and diagnostic approach to inflammatory cardiomyopathy. Inflammatory cardiomyopathy presents a high grade of clinical heterogeneity (clinical presentation may range from chronic heart failure to abrupt onset of life-threatening ventricular arrhythmias) and phenotypical diversity (non-invasive findings may mimic other cardiomyopathies such as ARVC, DCM, etc.). Autoimmune biomarkers may suggest an immune-mediated etiology and identify patients with worse phenotype and follow-up [35,36]. A diagnosis of clinically suspected myocarditis is mostly based on CMR findings, but only EMB can achieve a definitive and etiological diagnosis, possibly identifying candidates for tailored immunosuppression in virus-negative cases. ARVC: left ventricular cardiomyopathy; CAD: coronary artery disease; DCM: dilated cardiomyopathy; ECG: electrocardiogram; HCM: hypertrophic cardiomyopathy. Created with Biorender.

**Figure 2 cells-13-02082-f002:**
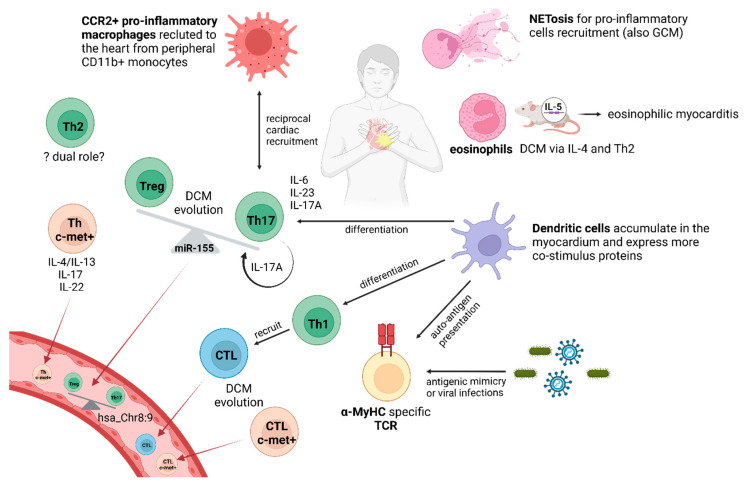
Summary of the role of cellular immunity in myocarditis within the inflamed heart and peripheral blood (bottom left). Myocarditis is the result of a fine interplay of different pro-inflammatory cells. α-MyHC: myosin heavy chain isoform α; CTL: cytotoxic T cell; DCM: dilated cardiomyopathy; GCM: giant cell myocarditis; IL: interleukin; TCR: T cell receptor; Th: T helper cell; Treg: T regulatory cell. Created with Biorender.

**Table 1 cells-13-02082-t001:** Cytokines and chemokines, listed in alphabetical order, that are relevant in myocarditis pathogenesis.

Cytokine	Role in Myocarditis
CCL5	Pro-inflammatory: CTL chemoattracting agent
IFN-γ	Pro-inflammatory: secreted by infiltrating T lymphocytes and increases cardiac tissue inflammation
IL-1β	Pro-inflammatory: important for innate immunity
IL-4	Pro-inflammatory: linked to Th2 cell differentiation
IL-5	Pro-inflammatory in eosinophilic myocarditis
IL-6	Pro-inflammatory: fundamental for myocarditis development and Th17 cell differentiation
IL-10	Anti-inflammatory: reduced in myocarditis
IL-17	Pro-inflammatory: secreted by Th17 cells and mediates myeloid cells recruitment, fibrosis, and favors DCM evolution
IL-22	Pro-inflammatory: secreted by Th22 cells
IL-23	Pro-inflammatory: fundamental for myocarditis development and Th17 cell differentiation
MIP-1α	Pro-inflammatory: macrophages and CTL chemoattracting agent
TGF-β1	Pro-inflammatory: favors Th17 cell differentiation
TNF-α	Pro-inflammatory

## Data Availability

Not applicable.

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
