# Peer review of "Cellular Immunology of Myocarditis: Lights and Shades—A Literature Review"

_cells, 2024, doi:10.3390/cells13242082_

Round 1
Reviewer 1 Report
Comments and Suggestions for Authors
This is a nice overview of the field.
1) It should be cleared that there are very few randomized controlled trials. This statement seems to be too strong. Line 252 refers to randomized clinical trials and meta-analyses but these are studies that are either not randomized, or open label, and should be interpreted with caution.
2) One randomized clinical trial is ARAMIS, and showed no benefit, but the conclusions about the trial do not seem supported. Line 458 refers to a likely involvement rather than auto-inflammatory mechanisms but there is no evidence to support this. ARAMIS showed that anakinra given for an average of 2 doses in low-risk patients had no significant effects, but event rates were very low, also there was a trend toward reduction in chest pain (not significant), therefore the role of anakinra in myocarditis is uncertain. Of note, several case series suggest a role for anakinra and the innate immunity in myocarditis (see Golino et al. - Int J Cardiol 2024).
Author Response
RESPONSE TO REVIEWER #1
This is a nice overview of the field.
OUR RESPONSE
Thank you for the appreciation for our work and for your comments, which we have addressed as follows.
- It should be cleared that there are very few randomized controlled trials. This statement seems to be too strong. Line 252 refers to randomized clinical trials and meta-analyses but these are studies that are either not randomized, or open label, and should be interpreted with caution.
OUR RESPONSE
We thank the Reviewer for his/her comment. We have blunted the sentence, and revised text reads as follows: “Evidence supporting the efficacy of IS for treating heart failure in lymphocytic virus-negative myocarditis mainly derives retrospective studies and meta-analyses”. (Lines 252-253)
- One randomized clinical trial is ARAMIS, and showed no benefit, but the conclusions about the trial do not seem supported. Line 458 refers to a likely involvement rather than auto-inflammatory mechanisms but there is no evidence to support this. ARAMIS showed that anakinra given for an average of 2 doses in low-risk patients had no significant effects, but event rates were very low, also there was a trend toward reduction in chest pain (not significant), therefore the role of anakinra in myocarditis is uncertain. Of note, several case series suggest a role for anakinra and the innate immunity in myocarditis (see Golino et al. - Int J Cardiol 2024).
OUR RESPONSE
We thank the Reviewer for his/her comment. According to this comment, we have deeply modified this part of the manuscript. Revised text reads as follows (lines 457-466):
“the ARAMIS (Anakinra versus placebo double-blind Randomized controlled trial for the treatment of Acute Myocarditis) phase 3 placebo-controlled trial did not show a net benefit in terms of in terms of heart failure episodes, chest pain, left ventricular ejection fraction <50%, and ventricular arrythmias in patients with clinically suspected acute myocarditis of unspecified etiology presenting with chest pain and normal left ventricular (LV) function [143]. This is likely due to the low adverse event rate in the trial, and to the prevalent involvement of autoimmune, rather than autoinflammatory mechanisms in a sizable proportion of myocarditis cases. Further studies are needed to clarify the role of IL-1 receptor antagonists in patients with biopsy-proven immune-mediated acute myocarditis, considering the evidence supporting the role of innate immunity in myocarditis (new reference: Golino et al, IJC, 2024)”.
Reviewer 2 Report
Comments and Suggestions for Authors
Authors have written an interesting revision about myocarditis, including origin/cause as well as cellular mechanism explaining in detail all phenotypic variables reported to date.
Minor points to clarify:
1.- Please, all genes should be written in italic.
2.- Despite explained in the text, a Table/Figure including pro-inflammatory/anti-inflammatory molecules which play a role in myocarditis should be of interest for readers.
3.- Additional data focused on cardiomyopathies (especially ACM entity, not only ARVC) should be of interest due to key role of inflammatory infiltrates in ACM diagnosis.
Author Response
RESPONSE TO REVIEWER #2
Authors have written an interesting revision about myocarditis, including origin/cause as well as cellular mechanism explaining in detail all phenotypic variables reported to date.
OUR RESPONSE
We appreciate the Revewer’s positive feedback on our work. We have carefully considered your comments and have addressed them in the following manner.
1.- Please, all genes should be written in italic.
OUR RESPONSE
Thank you for your comment. We have corrected the text and ensured that all genes name are written in Italics.This has been especially relevant in the section “Interplay of genetic predisposition and autoimmunity”, where the genes TTN and DSP were mentioned.
2.- Despite explained in the text, a Table/Figure including pro-inflammatory/anti-inflammatory molecules which play a role in myocarditis should be of interest for readers.
OUR RESPONSE
Thank you for your comment. We have added a Table in which the relevant pro-inflammatory/anti-inflammatory molecules (including cytokines and chemokines) have been briefly listed, to improve clarity. New Table 1 can be found at page 11.
3.- Additional data focused on cardiomyopathies (especially ACM entity, not only ARVC) should be of interest due to key role of inflammatory infiltrates in ACM diagnosis.
OUR RESPONSE
We thank the Reviewer for his/her comment. We have expanded the role of myocardial inflammation in the context of ACM. Revised text reads as follows:
- In a non-negligible proportion of rare cases… (line 80)
- This is even more relevant considering the newly identified phenotypes of arrhythmogenic cardiomyopathy (ACM), including the “left-dominant” and “biventricular” disease subtypes (new reference), in which a phenotypical and clinical overlap with inflammatory cardiomyopathy be carefully investigated, in order to promptly define a correct diagnostic and therapeutic patients’ work-up. Since histological evidence of inflammatory infiltrates in ARVC patients has been provided since the 90s, multi-parametric assessment of myocarditis in the context of ACM, especially during the so-called “hot phases”, is encouraged (new reference). (Lines 83-91)